# Kidney Bean Protein Prevents High-Fat and High-Fructose Diet-Induced Obesity, Cognitive Impairment, and Disruption of Gut Microbiota Composition

**DOI:** 10.3390/foods13111718

**Published:** 2024-05-30

**Authors:** Chunyang Jiang, Shiyu Li, Hang Su, Nong Zhou, Yang Yao

**Affiliations:** 1Key Laboratory of Grain Crop Genetic Resources Evaluation and Utilization, Ministry of Agriculture and Rural Affairs, Institute of Crop Sciences, Chinese Academy of Agricultural Sciences, No.12 Zhongguancun South Street, Haidian District, Beijing 100081, China; jh230612@163.com (C.J.); 82101222148@caas.cn (S.L.); suhang99077@163.com (H.S.); 2Chongqing Engineering Laboratory of Green Planting and Deep Processing of Famous-Region Drug in the Three Gorges Reservoir Region, College of Biology and Food Engineering, Chongqing Three Gorges University, Chongqing 404120, China; erhaizn@126.com

**Keywords:** kidney bean protein, obesity, cognitive impairment, gut microbiota

## Abstract

A long-term intake of a high-fat and high-fructose diet (HFFD), even a high-fat, high-fructose but low-protein diet (HFFD + LP), could cause obesity associated with cognitive impairments. In the present study, rats were subjected to a normal diet (ND), an HFFD diet, an HFFD + LP diet, and an HFFD with kidney bean protein (KP) diet for 8 weeks to evaluate the effect of KP on HFFD- or HFFD + LP-induced obesity and cognitive impairment. The results demonstrated that compared with the HFFD diet, KP administration significantly decreased the body weight by 7.7% and the serum Angiotensin-Converting Enzyme 2 (ACE-2) and Insulin-like Growth Factor 1 (IGF-1) levels by 14.4% and 46.8%, respectively (*p* < 0.05). In addition, KP suppressed HFFD-induced cognitive impairment, which was evidenced by 8.7% less time required to pass the water maze test. The 16s RNA analysis of the colonic contents showed that the relative abundance of *Bifidobacterium*, *Butyricimonas*, and *Alloprevotella* was increased by KP by 5.9, 44.2, and 79.2 times. Additionally, KP supplementation primarily affected the choline metabolic pathway in the liver, and the synthesis and functional pathway of neurotransmitters in the brain, thereby improving obesity and cognitive function in rats.

## 1. Introduction

The consumption of high-calorie foods and a lack of physical activity, among other lifestyle changes, are the primary contributors to obesity. According to the World Obesity Atlas 2023, the global population of obese children has reached 179 million, and it is predicted to rise to 241 million by 2025, accounting for 38% of the total global child population. Previous studies showed that the long-term intake of a high-fat and high-fructose diet (HFFD) or HFFD and low protein (LP) diet was associated with the occurrence of obesity and cognitive impairments [1,2]. Mounting evidence suggested that alterations in the composition and diversity of gut microbiota, termed dysbiosis, significantly correlate with the development and progression of both obesity and cognitive dysfunction. Obesity is a generalized low-grade inflammatory state [3,4]. Moreover, experimental studies in animal models demonstrated that the transplantation of gut microbiota from obese individuals into germ-free rats induced weight gain and metabolic disturbances, further implicating the role of gut microbiota in obesity pathogenesis. Specifically, pathogenic Gram-negative bacteria in gut microbiota played a pivotal role in obesity by instigating pro-inflammatory signaling cascades through their interaction with toll-like receptors (TLRs) in the mucosa and peripheral tissues [5]. Conversely, bacteria from the *Bacteroidetes* phylum can enhance the differentiation of regulatory T cells, thus mitigating inflammatory responses and potentially averting the development of obesity [6]. Concomitantly, accumulating evidence supports the notion that gut microbiota dysbiosis may contribute to cognitive impairments, including deficits in learning and memory functions. Dysregulated gut–brain axis communication, mediated by microbial metabolites, inflammatory mediators, and neural signaling pathways, has been proposed as a plausible mechanism linking gut microbiota dysbiosis to cognitive dysfunction. In a cross-sectional study, the abundance of *Bacteroidetes* in the gut microbiota was found to be lower in patients with cognitive impairment compared with normal subjects [7]. At the species level, the richness of the *Prevotellaceae* was notably deficient in those suffering from cognitive impairments [8]. Therefore, the target modulation of gut microbiota could be an alternative strategy for preventing obesity and accompanying cognitive impairment.

Dietary intervention with plant protein has been demonstrated to be an effective strategy to prevent obesity and cognitive impairment by modulating gut microbiota. Previous research proposed that dietary interventions incorporating legumes can protect the cognitive function and skeletal and immune systems of malnourished children in impoverished regions [9]. Our previous study demonstrated that both chickpea protein and mung bean protein can upregulate the diversity of gut microbiota in rats, hence ameliorating cognitive impairment induced by a low casein diet [10]. Our study also demonstrated that kidney bean protein (KP), possessing alpha-amylase inhibitor activity, can hinder the breakdown of starch in food into sugars, thus resulting in more carbohydrates entering the gut, ultimately modulating body weight by increasing the ratio of *Bacteroidetes*/*Firmicutes* in colonic content [11]. Nonetheless, the direct impact of kidney bean protein on HFFD-induced obesity and cognitive impairment is yet to be thoroughly investigated.

Therefore, we conducted a dietary intervention rich in kidney bean protein using a rat model to (1) investigate the anti-obesity and cognitive impairment prevention effect of kidney bean protein on HFFD-induced and HFFD + LP-induced obese rats, (2) determine whether KP attenuates cognitive impairment by regulating the gut microbiota, and (3) discuss the influence of KP on metabolism.

## 2. Materials and Methods

### 2.1. Animal Treatment

A total of 32 male Sprague Dawley (SD) rats in mid-weaning, 4 weeks of age (90.77 ± 5.21 g), were provided by Beijing Vital River Laboratory Animal Technology Co., Ltd. After a week of acclimatization and maternal feeding, the rats were divided evenly into four groups as follows: (1) the normal diet group (ND); (2) the HFFD group feed with a high-fat and high-fructose diet; (3) the HFFD + LP group feed with a low casein of HFFD; (4) the HFFD + KP group feed with a HFFD with KP (Table 1). The experimental design was based on that of Brown et al. 2015 [12]. Throughout the experiment, the rats had ad libitum access to food and water. All rats were housed under controlled conditions at 24 °C, with a 12 h light/dark cycle. Body weight and tail length were measured every two weeks according to the Diana et al. 2020 [13] method for a total feeding period of 8 weeks (Figure 1). On day 56, after 12 h of fasting, all groups of rats were euthanized under ether anesthesia, blood was collected from the inferior vena cava, and serum was immediately separated. The levels of IGF-1 and ACE-2 in the serum were detected using ELISA kits. Brain and liver tissues were also collected. All samples were stored at −80 °C until analysis. This study was conducted in accordance with the European Community Guidelines for the Use of Experimental Animals and was approved by the Pony Testing International Group’s Animal Care and Use Committee.

### 2.2. Raw Materials

ELISA kits for insulin-like growth factor-1 (IGF-1) were commercially available from Abcam (Cambridge, UK). An ELISA kit for angiotensin-converting enzyme-2 (ACE-2) was procured from Biorbyt Limited (Cambridgeshire, UK). The E.N.Z.A ^®^ Stool DNA Kit was supplied by Omega Bio-Tek Inc. (Norcross, GA, USA). All other chemicals were of analytical reagent grade. Kidney bean protein (purity, ≥85.00%) was obtained from Gushen Biological Technology Group Co., Ltd. (Dezhou, China).

### 2.3. Morris Water-Maze Tests

The positioning cruise experiment was performed as previously reported by Liu et al. 2020 [14]. A Morris water maze round swimming pool (150 cm in diameter and 60 cm in height) was used. The pool was evenly divided into fourths, creating four quadrants of the same area. During the experiment, the temperature of the water was maintained at 23 °C. A visual platform (4.5 cm in diameter and 25 cm high) was placed 1.5 cm below the water surface in each quadrant. Rats were randomly placed in any position among these four quadrants, and the time it took them to locate the platform was recorded. If a rat was unable to find the platform within 60 s, it was guided to the platform using a guide pole and allowed to remain on the platform for 30 s. This process was repeated in all four quadrants. A video surveillance software 1.1.2 (Super maze, Shanghai, China) system was used to record the rats’ swimming trajectories and the time it took to locate the platform. After four days of training, a space exploration experiment was conducted with the platform removed. The rats’ movement trajectory within 60 s, the number of times they crossed the original platform location, and their stays in the target quadrant were all recorded.

### 2.4. Steps to Obtain 16S rRNA Gene Sequences

Colon samples were collected and prepared using the E.N.Z.A ^®^ fecal DNA isolation kit, following the instructions provided by the manufacturer. DNA extraction was carried out using the forward primer 5′-ACTCCTACGGGAGGCAGCA-3′ and reverse primer 5′-GGACTACHVGGGTWTCTAAT-3′ targeting the V3-V4 hyper-variable region. The microbiome genomic DNA samples and raw data were sequenced and analyzed on the Illumina HiSeq 2500 platform (Illumina, San Diego, CA, USA). Operational Taxonomic Units (OTUs) were clustered based on 97% similarity, according to the Greengenes database (V.13.8). Relative abundance was calculated at the phylum, order, and genus levels. Differences were analyzed using the α-diversity index (Shannon index).

### 2.5. Untargeted Metabolomics of the Liver

Liver samples (25 mg) were treated with 200 μL of water and 480 μL of extraction solvent (methyl tert-butyl ether–methanol = 5:1), vortexed for 30 s, ground at 35 Hz for 4 min, and sonicated in ice water for 5 min The grinding and sonication steps were repeated twice. The samples were allowed to stand at −40 °C for 1 h and then centrifuged at 3000 rpm/min (4 °C, 15 min); the supernatant was transferred (300 μL) and vacuum dried. Reconstitution was performed by the addition of 100 μL of the solution (dichloromethane–methanol = 1:1), vortexed for 30 s, sonicated for 10 min in an ice water bath, centrifuged at 13,000 rpm/min for 15 min, and then the supernatant was collected for testing. The liver metabolites were analyzed using the following method: An Agilent 1290 (Agilent Technologies, Santa Clara, CA, USA) UHPLC system was used with a chromatographic column of Phenomen Kinetex C18 (Santa Clara, CA, USA) (2.1 × 100 mm, 1.7 μm) and a Triple TOF 6600 Mass spectrometer (AB SCIEX, Redwood City, CA, USA). The mobile phases were water–acetonitrile (4:6) with 10 mmol/L ammonium formate (A) and acetonitrile–isopropanol (1:9) with 10 mmol/L ammonium formate solution added to 1000 mL of 50 mL (B). The elution gradient was 0~12.0 min, 40~100% B; 12.0~13.5 min, 100% B; 13.5~13.7 min, 100~40% B; and 13.7~18.0 min, 40% B. The mobile phase flow rate was 0.3 mL/min, the column temperature was 45 °C, the sample temperature was 4 °C, and the injection volume was 1 μL for the positive ion mode and 2 μL for the negative ion mode. The collisionally induced dissociation energy was 45 eV. The ion source parameters were as follows: GS1: 60 psi, GS2: 60 psi, CUR: 30 psi, TEM: 600 °C, DP: 100 V, and ISVF: 5000 V (Pos)/−4500 (Neg).

### 2.6. Targeted Metabolomics of Brain Tissues

Brain tissue samples (1 mg) were treated with 20 μL H_2_O and 80 μL pre-chilled (−20 °C acetonitrile, 0.1% FA) for extraction. The sample was vortexed for 30 s, homogenized (45 Hz, 4 min), and sonicated in ice water for 5 min, repeated 3 times, followed by an overnight incubation at −40 °C. The sample was then centrifuged (12,000× *g*, 4 °C, 15 min), and 80 μL of the supernatant was mixed with 40 μL of 100 mmol/L sodium carbonate solution, and 40 μL of 2% phenacyl chloride acetonitrile solution and incubated for 30 min. Afterward, 10 μL of internal standard was added and centrifuged (12,000× *g*, 15 min, 4 °C). The supernatant (40 μL) was mixed with 20 μL of H_2_O and reserved for analysis. The analysis was conducted using ultra-high performance liquid chromatography–tandem mass spectrometry (UHPLC-MS/MS). The chromatographic column Waters ACQUITY UPLC HSS T3 (100 × 2.1 mm, 1.8 μm) was installed on the Exion LC system (Waters Corporation, Milford, MA, USA). The mass spectrometer was AB Sciex QTrap 6600 (AB SCIEX, Redwood City, CA, USA). The mobile phases were 0.1% formic acid and 1 mm/L ammonium formate aqueous solution (A) and acetonitrile (B). The column temperature was 40 °C, and the sample temperature was maintained at 6 °C with an injection volume of 1 μL. The typical ion source parameters were as follows: ion spray voltage: +5000 V, curtain gas: 35 psi, temperature: 400 °C, ion source gas1: 60 psi, and ion source gas2: 60 psi. Multiple reaction monitoring (MRM) data were processed by Skyline software 20.2 (Redwood City, CA, USA).

### 2.7. Data Analysis

For the results of this study, data were quantified and reported as mean ± SD. For significant differences between mean values, one-way ANOVA was performed using IBM SPSS Statistics 20 (SPSS Inc. Chicago, IL, USA). The Tukey test was performed for multiple comparisons. The mean was considered to be significantly different at *p* < 0.05. The apparent indicator data and bacterial sequencing by 16S rRNA were imported into the R package 4.3.1 (Vienna, Austria). Then, the correlation and *p*-value were calculated using the cor.test., and the output data were drawn using ggplot Graph software 3.3.0 (Springer-Verlag, New York, NY, USA).

## 3. Results and Discussion

### 3.1. Effect of Supplementary Kidney Bean Protein on the Growth Profile of Obese Rats

At 8 weeks, compared with the HFFD group, rats fed with ND, HFFD + LP, and HFFD + KP diets showed weight reductions of 9.1%, 6.3%, and 7.7%, respectively (Table 2). This revealed that the inclusion of kidney bean protein in the rat diet suppressed weight gain induced by a high-fat high-fructose diet (*p* < 0.05). It is noteworthy that in the eighth week, compared with the ND group, there was no significant difference in the body weight of rats in the HFFD + LP and HFFD + KP groups. Although the HFFD + LP diet contained more fat and sugar, the content of protein in the diet was reduced. The reduction in protein intake led to malnutrition, limiting the growth of rats. The reason why the body weight of the rats in the HFFD + KP group showed no significant difference from the ND group could be that the kidney bean protein contains functional proteins such as alpha-amylase inhibitors, which had a significant inhibitory effect on the weight gain of obese rats [15,16]. At 8 weeks, compared with the HFFD + LP group, rats fed with the ND, HFFD, and HFFD + KP groups showed tail length increases of 11.5%, 14.0%, and 10.6%, respectively (*p* < 0.05) (Table 2). Weight-to-length ratio measurements are often used to assess the health status of children [17]; hence, the body weight/tail length ratio of the rats was examined. Compared with the HFFD group, rats fed with the ND and HFFD + KP diets showed a reduction in the body weight/tail length ratio by 14.2% and 11.9%, respectively (*p* < 0.05); however, the body weight/tail length ratio surprisingly increased by 8.8% in rats fed with the HFFD + LP diet (*p* < 0.05) (Table 2). This suggested that long-term consumption of high-fat, high-fructose, and low-protein food can lead not only to obesity but also to malnutrition. In addition, at 8 weeks, compared with the ND group, there was no significant difference in the weight/tail length ratio of the rats in the HFFD + KP group. This suggests that the rats were consuming an adequate amount of protein, their growth was balanced, and there was no occurrence of malnutrition (Table 2). In conclusion, adding kidney bean protein to the diet can significantly mitigate issues of obesity and malnutrition in rats. Nonetheless, it is essential to note that in this experimental design, all dietary interventions were conducted concurrently. Therefore, we can only assess the preventive effects of KP on obesity and memory impairments induced by a high-fat and high-lactose diet. In subsequent research, it is necessary to evaluate the treatment effect of KP on established models of obesity and malnutrition.

IGF-1 and ACE-2 are biomarkers significantly associated with cell growth, cell repair, blood pressure regulation, and glucose metabolism. After 8 weeks of feeding, as compared with the HFFD group, there was a striking reduction of 14.4% and 46.8% in the levels of IGF-1 and ACE-2, respectively, in the HFFD + LP group rats (Figure 2A,B), indicating a significant impact on the growth status of the rats, in harmony with previous results on the tail length and the body weight/tail length ratio. Similarly, compared with malnourished rats, supplementation of chickpea protein and mung bean protein in the diet significantly elevated the levels of IGF-1 and ACE-2 in the body [10,17]. Comparatively, there was no significant difference in the levels of IGF-1 and ACE-2 in the HFFD + KP group, which was closest to the levels in the normal diet group, suggesting the beneficial role of kidney bean protein in improving the health status of obese and malnourished rats. IGF-1 plays a pivotal role in brain development and is associated with promoting neuron differentiation and growth. A significant decrease in IGF-1 levels could potentially disrupt neuronal development and connectivity [18]. Following a four-day place navigation test, a change was observed in the escape latency of all rat groups (Figure 2C). The time taken to find the visual platform increased by 143% in the HFFD + LP group compared with the HFFD group, whereas the time was reduced by 8.7% in rats whose diet was supplemented with kidney bean protein, correlating with the previously mentioned notable decrease in IGF-1 levels in the HFFD + LP rats (Table 3). The distance traveled by the HFFD + LP rats in the target quadrant to find the visual platform increased by 163% compared with the HFFD group, while the distance traveled to find the platform among rats fed with kidney bean protein remained statistically indistinguishable (Figure 2D). This suggested that supplementing the diet with kidney bean protein can considerably ameliorate cognitive impairment and spatial cognitive ability in obese rats.

### 3.2. Gut Microbiota Composition in Rats

Changes in the gut microbiota in each group were detected using 16S rRNA sequencing. Compared with the HFFD + LP group, the OUT and Shannon index of the HFFD + KP group increased by 20.1% and 18.0%, respectively (Figure 3A,B). Compared with the ND group, the OUT and Shannon Index also showed a significant increase. This indicated that the gut microbiota of rats in the HFFD + KP group exhibited superior species abundance and uniformity. Previous reports have also confirmed significant changes in the abundance and diversity of the gut microbiota in rats fed a high-fat and high-fructose diet [19]. At the phylum level, *Firmicutes*, *Bacteroidetes*, and *Proteobacteria* were dominant in each group, but the relative proportions differed among the groups. Compared with the HFFD + LP and ND groups, the HFFD + KP group showed a reduction of 32.3% and 33.5% in *Firmicutes* and 50.3% and 58.6% in *Bacteroidetes* and an increase of 229.5% and 310.9% in *Proteobacteria* (Figure 3C). At the order level, *Clostridiales* was dominant in each group. Compared with the HFFD + LP group, the HFFD + KP group showed a decrease of 27.2% in *Clostridiales* and 71.8% in *Bacteroidales*, as well as an increase of 123.4% in *Lactobacillales* (Figure 3D). These changes indicated that the gut microbiota was modulated to a healthier state. Compared with the HFFD + LP and ND groups, *Lactobacillus* in the HFFD + KP group significantly increased by 15.8 times and 1.2 times (Figure 3E). *Lactobacillus* is a strain of microorganisms that has been extensively studied and is beneficial for neuroinflammatory-related diseases. *Lactobacillus* effectively protected against memory deficits and Alzheimer’s disease in aging and neuroinflammatory mouse models [20,21]. Compared with the HFFD + LP group, *Bifidobacterium* in the HFFD + KP group increased by 5.9 times. *Bifidobacterium* is an “immune biotic” that beneficially modulates neuroinflammatory responses and behavior in many model neuroinflammatory-related diseases [22]. Compared with the HFFD + LP group, *Butyricimonas* and *Alloprevotella* in the HFFD + KP group increased by 44.2 and 79.2 times (Figure 3E). These bacteria were considered to be associated with the production of SCFAs (Figure 3E). In addition, the relative abundance of the healthy gut-dominant bacterial group *Ruminococcaceae_UCG-005* in the HFFD + KP group was 54.8 times that of the HFFD + LP group. Compared with the HFFD + LP group, the related abundance of *Lachnospiraceae* in the HFFD + KP group decreased by 5.37 times. This bacterium was significantly related to the production of short-chain fatty acids. A decrease in *Lachnospiraceae* abundance might reduce the production of acetic acid, thereby reducing liver inflammation and fat deposition, further improving liver function, and promoting glucose metabolism [23]. In addition, in a high-energy diet-induced cognitive impairment rat model, *Lachnospiraceae* negatively correlated with location recognition memory. Similarly, the relative abundance of *Enterococcus* and *Ruminococcus* torques in the HFFD + KP group was significantly reduced. All in all, these findings suggested that high-fructose diet intestinal damage was transformed into a healthier gut following dietary intake of kidney bean protein.

### 3.3. Liver Non-Targeted Metabolomics Analysis

Liver metabolites were detected using an LC-MS/MS coupled quadrupole orbitrap mass spectrometer. The results of liver lipid metabolites showed that the choline metabolic pathway is the main pathway involved under the intervention of a kidney bean protein diet. In the HFFD + KP group, compared with the HFFD + LP group, 251 types of differential metabolites that match the secondary mass spectrometry type were identified (Figure 4A). Compared with the ND group, 162 differential metabolites were detected. The main metabolite that underwent changes was phosphatidylcholine (Figure 4C). Phosphatidylcholine is a major precursor in the choline metabolic pathway, which can be broken down into choline and then transformed into acetylcholine under the action of enzymes. Enhancing the level of phosphatidylcholine could improve the memory ability of rats [24]. Phosphatidylserine was also significantly upregulated. Phosphatidylserine is also an important metabolite in the choline metabolic pathway. The upregulation of phosphatidylserine could increase the synthesis of phospholipids, further promote the production of choline, enhance the connection of information transmission between cells, and improve the efficiency of signal transmission in neurons during the learning and memory process, thereby enhancing memory ability [25].

In comparison with the HFFD and ND groups, a total of 214 and 173 differential metabolites were identified in the HFFD + KP group (Figure 4B). The main substance that is downregulated in HFFD is phosphatidylcholine (Figure 4D). In the choline nutritional metabolic pathway in the liver, phosphatidylcholine is primarily used for the production of bile, which aids in the digestion of fats in the small intestine. A reduction in phosphatidylcholine levels might affect the production of bile, thereby influencing the breakdown and absorption of fats. This could lead to an excessive accumulation of fat in the body, increasing the risk of obesity [26].

### 3.4. Brain-Targeted Metabolomics Analysis

Brain metabolites were detected using an LC-MS/MS coupled quadrupole orbitrap mass spectrometer. In the HFFD + KP group, compared with the HFFD + LP group, 404 types of differential metabolites were identified. When compared with the HFFD group, a total of 366 differential metabolites were found (Figure 5A,B). The kidney protein diet intervention primarily affected the neurotransmitter synthesis and function pathways and phosphocreatine energy metabolism pathway in rats. In the HFFD + KP group, the levels of D-alamine, L-glutamine, and L-serine were 1.27, 1.58, and 1.48 times those of the HFFD + LP group, respectively. They were 1.22, 1.52, and 1.70 times those of the ND group, respectively. These types of amino acids indirectly influence the memory function of organisms by affecting neurotransmitters and the transmission of neural signals [27,28]. The creatine level in the HFFD + KP group was 1.48 and 1.20 times higher than that in the HFFD + LP and ND groups. When learning new information or forming new memories, the signal transmission between neurons in the brain could be enhanced. To a certain extent, creatine could efficiently conduct neural signals [29]. The level of hypoxanthine in the HFFD + KP group was 0.54 and 0.84 times that of the HFFD + LP and ND groups. Hypoxanthine is known to inhibit the activity of Na^+^, K^+^-ATPase in rat striatal synaptic membranes, thereby affecting the transmission of nerve impulses, and its accumulation is considered one of the causes of neurological disorders [30]. The primary function of creatine is to supply energy to neurons. Norvaline was found to increase the levels of nitric oxide synthase, which in turn reduced the permeability of the blood–brain barrier, decreased the proliferation of microglial cells, and astroglial degeneration. This improved cognitive impairments and memory function in organisms [24,31]. Compared with HFFD + LP, HFFD + KP significantly increased the levels of norvaline within the mouse brain (Figure 5C).

### 3.5. Analysis of the Correlation among Gut Microbiota, Hepatic Lipid Metabolites, and Brain Metabolites

Figure 6 shows the Mantel test results of rat intestinal gut microbiota, brain metabolites, and hepatic lipid metabolites. The illustration clearly delineates that in the context of hepatic lipid metabolites and gut microbial metabolomics in rats, there is a significant positive correlation (*p* < 0.05) between sphingolipids (Sph) and the abundance of gut microbes such as *Bacteroidetes*, *Verrucomicrobia*, and *Tenericutes*; likewise, glucosylceramides (GlcCers) show a significant positive correlation (*p* < 0.05) with *Firmicutes* and *Actinobacteria*. In terms of brain metabolites and gut microbial metabolite analyses in rats, organoheterocyclic compounds are significantly positively correlated (*p* < 0.05) with *Verrucomicrobia* and *Proteobacteria*. Considering the analysis between rat brain metabolites and hepatic metabolites, phosphatidylinositols (PIs) exhibit a significant positive correlation (*p* < 0.05) with organic oxygen compounds and organooxygen compounds and a significant negative correlation (*p* < 0.05) with organic compounds. These results suggest that the alterations in gut microbiota induced by KP are likely the primary factor causing changes in serum and brain metabolites, subsequently improving obesity and memory impairments. In further studies, to confirm this conclusion, fecal microbiota transplantation experiments in germ-free rats are imperative. This will also be the focus of our follow-up studies.

## 4. Conclusions

In conclusion, the intake of kidney bean protein had a positive impact on the physiological status of obese rats induced by the HFFD, as evidenced by improved body weight, tail length, and serum biomarkers. The 16S rRNA sequencing results showed that *Bifidobacterium*, *Butyricimonas*, and *Alloprevotella* in the HFFD + KP group were significantly improved in relative abundance compared with the HFFD + LP group. Non-targeted metabolomics revealed that supplementary kidney protein primarily affected the choline metabolic pathway in the liver, as well as the synthesis and functional pathways of neurotransmitters in the brain. It upregulated the levels of phosphatidylcholine and neurotransmitters in the brain, thereby improving obesity and cognitive function in rats. In this study, we reported for the first time the potential of kidney protein to prevent cognitive impairment, which was evidenced by changes in gut microbial groups, liver metabolites, and brain metabolites.

## Figures and Tables

**Figure 1 foods-13-01718-f001:**
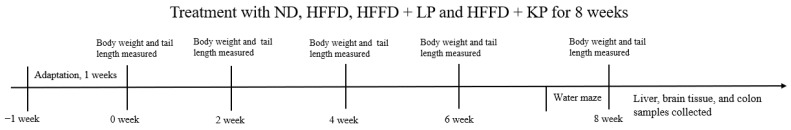
Methodology scheme.

**Figure 2 foods-13-01718-f002:**
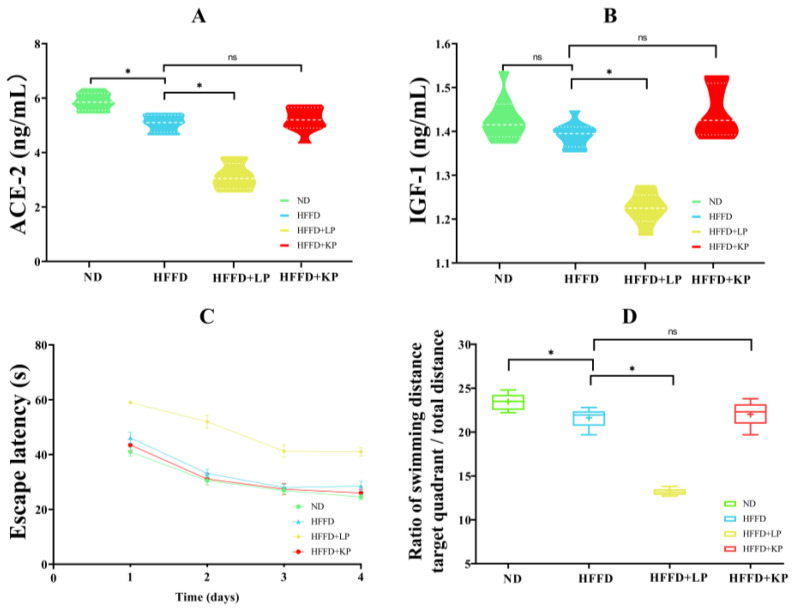
(**A**) ACE-2; (**B**) IGF-1; (**C**) escape latency; and (**D**) the ratio of swimming distance target quadrant/total distance. Abbreviations: (1) ND, normal diet; (2) HFFD; (3) HFFD + LP; (4) HFFD + KP. Data are expressed as mean ± SD, *n* = 8, ns = not significant, * *p* < 0.05.

**Figure 3 foods-13-01718-f003:**
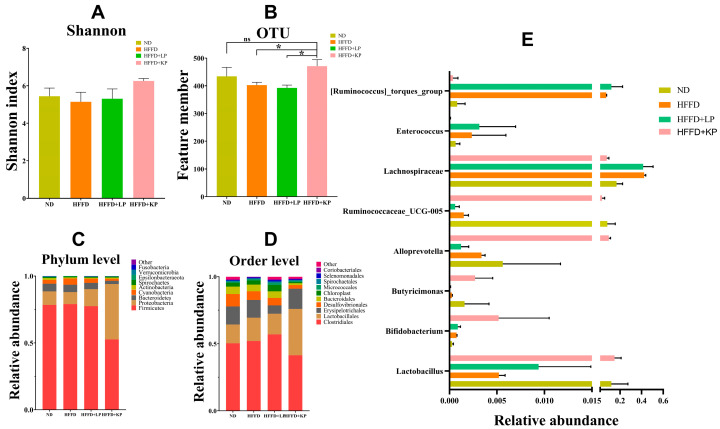
(**A**) The Shannon index among groups; (**B**) the OTU among groups; (**C**) the gut microbiota at the phylum level among groups; (**D**) the gut microbiota at the order level; and (**E**) the relative abundance of *Lactobacillus*, *Bifidobacterium*, *Butyricimonas*, *Ruminococcaceae_UCG-005*, *Lachnospiraceae*, *Alloprevotella*, *Enterococcus*, and *Ruminococcus_torques_group* among groups. Data are expressed as mean ± SD, *n* = 8, ns = not significant, * *p* < 0.05 compared with HFFD + KP.

**Figure 4 foods-13-01718-f004:**
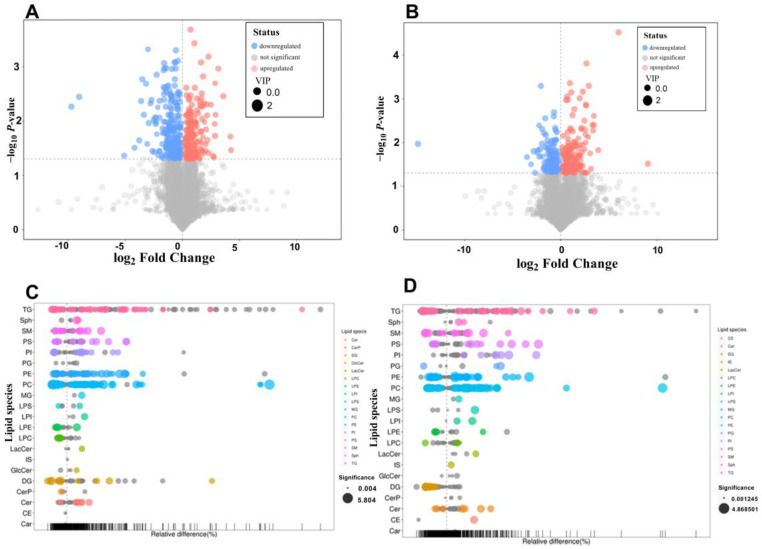
(**A**) Volcano plot for the HFFD + KP group vs. the HFFD + LP group; (**B**) volcano plot for the HFFD + KP group vs. the HFFD group; (**C**) bubble plot for the HFFD + KP group vs. the HFFD + LP group; and (**D**) bubble plot for the HFFD + KP group vs. the HFFD group. Data are expressed as mean ± SD, *n* = 8.

**Figure 5 foods-13-01718-f005:**
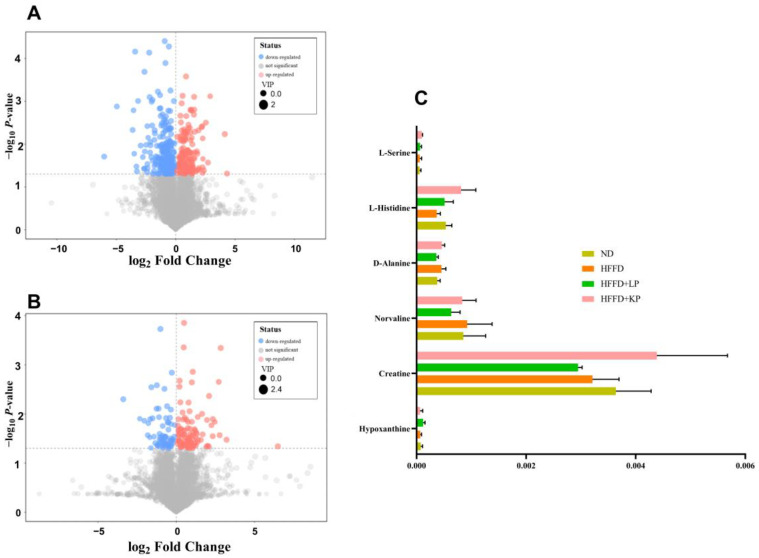
(**A**) Volcano plot for the HFFD + KP group vs. the HFFD + LP group; (**B**) volcano plot for the HFFD + KP group vs. the HFFD group; and (**C**) the relative content of hypoxanthine, creatine, norvaline, D-alanine, L-histidine, and L-serine among groups. Data are expressed as mean ± SD, *n* = 8.

**Figure 6 foods-13-01718-f006:**
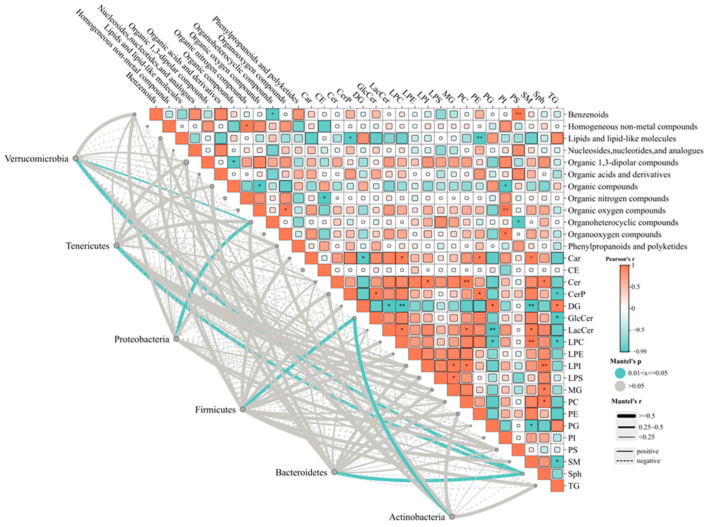
The Mantel test results of rat intestinal gut microbiota, brain metabolites, and hepatic lipid metabolites. The size of the square is proportional to the absolute value of the correlation coefficient. The line color and thickness represent the *p*-value and R value of the Mantel test.

**Table 1 foods-13-01718-t001:** Composition of diets.

Component	ND	HFFD	HFFD + LP	HFFD + KP
Casein/g	200	200	70	-
L-cystine/g	3	3	3	3
Kidney bean protein/g	-	-	-	200
Corn starch/g	397.5	217.5	347.5	217.5
Maltodextrin 10/g	132	132	132	132
Sucrose/g	100	100	100	100
Cellulose, BW200/g	50	50	50	50
Lard/g	70	250	250	250
Vitamin mix V10037/g	10	10	10	10
Mineral mix S10022G/g	35	35	35	35
Choline bitartrate/g	2.5	2.5	2.5	2.5
Energy supply ratio of each nutrition component				
Protein (%)	19	18	6	18
Carbohydrate (%)	65	37	49	37
Fat (%)	15	45	45	45

The abbreviations are as follows: normal diet (ND), high-fat and high-fructose diet (HFFD), high-fat, high-fructose but low protein diet (HFFD + LP), and kidney bean protein diet (HFFD + KP).

**Table 2 foods-13-01718-t002:** Rat body weight, tail length, and body weight-to-tail length ratio over eight weeks.

	Group	Week 0	Week 2	Week 4	Week 6	Week 8
Body weight (g)	ND	77.84 ± 3.47 b	172.65 ± 8.57 a	236.93 ± 5.65 b	317.04 ± 8.10 c	391.03 ± 7.95 b
	HFFD	85.11 ± 4.60 a	180.76 ± 6.87 a	250.64 ± 10.22 a	344.14 ± 8.8 a	430.28 ± 14.92 a
	HFFD + LP	84.46 ± 6.58 ab	176.75 ± 10.74 a	244.61 ± 9.90 ab	337.14 ± 12.45 ab	403.09 ± 11.69 b
	HFFD + KP	85.55 ± 5.60 a	177.49 ± 7.27 a	243.94 ± 7.61 ab	328.60 ± 8.53 bc	397.18 ± 8.11 b
Tail length (cm)	ND	9.71 ± 0.57 a	13.18 ± 0.84 a	14.80 ± 0.75 b	16.31 ± 0.69 ab	18.23 ± 0.80 a
	HFFD	10.19 ± 0.57 a	13.49 ± 1.20 a	15.33 ± 1.15 a	17.03 ± 1.02 a	18.75 ± 1.08 a
	HFFD + LP	10.39 ± 0.20 a	13.01 ± 0.77 a	14.08 ± 0.87 ab	15.40 ± 0.73 b	16.13 ± 0.73 b
	HFFD + KP	11.45 ± 3.39 a	13.34 ± 0.72 a	14.95 ± 0.67 ab	16.48 ± 0.59 a	18.05 ± 0.84 a
Body weight/Tail length	ND	8.03 ± 0.40 a	13.12 ± 0.44 a	16.04 ± 0.67 b	19.46 ± 0.65 b	21.48 ± 0.73 c
	HFFD	8.36 ± 0.26 a	13.47 ± 0.81 a	16.41 ± 0.83 b	20.27 ± 0.98 b	23.00 ± 1.05 a
	HFFD + LP	8.13 ± 0.55 a	13.60 ± 0.69 a	17.42 ± 0.79 a	21.93 ± 1.07 a	25.03 ± 0.95 b
	HFFD + KP	7.89 ± 1.51 a	13.30 ± 0.44 a	16.23 ± 0.56 b	19.86 ± 0.50 b	21.96 ± 0.98 bc

Abbreviations: (1) ND, normal diet; (2) HFFD, high-fat and high-fructose diet; (3) HFFD + LP, high-fat, high-fructose but low protein diet; (4) HFFD + KP, high-fat, high-fructose and kidney bean protein diet. The results are expressed as mean ± SD (*n* = 8), and different letters in the same column indicate significant differences (*p* < 0.05).

**Table 3 foods-13-01718-t003:** The escape latency of rats in four days.

Group	1	2	3	4
ND	41.00 ± 1.41 d	30.50 ± 1.50 c	26.88 ± 1.27 b	24.50 ± 0.87 c
HFFD	46.13 ± 1.83 b	33.13 ± 1.45 b	28.00 ± 1.12 b	28.50 ± 1.66 b
HFFD + LP	59.00 ± 0.71 a	52.00 ± 2.18 a	41.25 ± 2.11 a	41.00 ± 1.41 a
HFFD + KP	43.5 ± 1.87 c	31.13 ± 2.03 bc	27.38 ± 1.87 b	26.00 ± 1.22 c

The results are expressed as mean ± SD (*n* = 8), and different letters in the same column indicate significant differences (*p* < 0.05).

## Data Availability

The original contributions presented in the study are included in the article, further inquiries can be directed to the corresponding author.

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
