# Peer review of "Kidney Bean Protein Prevents High-Fat and High-Fructose Diet-Induced Obesity, Cognitive Impairment, and Disruption of Gut Microbiota Composition"

_foods, 2024, doi:10.3390/foods13111718_

Round 1

Reviewer 1 Report

Comments and Suggestions for Authors

Title : Kidney Bean Protein Suppresses High-Fat and High-Fructose 2 Diet-Induced Obesity and Cognitive Impairment in Rats by 3 Modulation of Gut Microbiota

This research assesses the impact of consuming bean protein at a 20% dose on obesity induced by a high-fat, high-fructose diet (HFFD). The parameters under examination include body weight, tail length, and levels of IGF-1 and ACE-2. In my view, this iteration of the manuscript necessitates several elucidations.

1.     The abstract lacks specificity, omitting numerical values and failing to highlight the significance of the observed variations. Out of the 12 lines, only 5 are dedicated to describing the results, and they are presented in a overly general manner.

2.     Lines 36-40 of the introduction provide brief information without referencing sources.

3.     Regarding line 90, the notion of administering fat in water seems perplexing as fat is not water-soluble. Additionally, there might be concerns about ensuring each rat receives the appropriate dose, especially when considering the typical method of incorporating fructose and fat into the diet.

4.     Furthermore, it's essential to understand the rationale behind selecting the 20% dose of bean protein.

5.     The duration of treatment for 56 days may raise questions, given the well-established understanding that obesity typically develops over a longer period with an HFFD, typically at least three months.

Author Response

Response to Reviewer 1 Comments

Point 1 The abstract lacks specificity, omitting numerical values and failing to highlight the significance of the observed variations. Out of the 12 lines, only 5 are dedicated to describing the results, and they are presented in a overly general manner.

Response 1 Thank you very much for your comments. I revised the abstract to make it more specific.

Abstract: A long-term intake of a high-fat and high-fructose diet (HFFD), even high-fat, high-fructose but low protein diet (HFFD+LP), could cause obesity associated with cognitive impairments. In the present study, rats were respectively subjected to a normal diet (ND), a HFFD diet, a HFFD+LP diet and a HFFD with kidney bean protein (KP) diet for 8-week, to evaluate the effect of KP on the HFFD or HFFD+LP induced obesity and cognitive impairment. The results demonstrated that compared to the HFFD diet, KP administration significantly decreased the body weight by 7.7%, and the serum Angiotensin-Converting Enzyme 2 (ACE-2) and Insulin-like Growth Factor 1 (IGF-1) levels by 14.4% and 46.8% respectively (p<0.05). In addition, KP suppressed the HFFD-induced cognitive impairment, which was evidenced by an 8.7% less time to pass the water maze test. The 16s RNA analysis of the colonic contents showed that the relative abundance of Bifidobacterium, Butyricimonas, and Alloprevotella were increased by KP by 5.9, 44.2, and 79.2 times. Additionally, KP supplementation primarily affected the choline metabolic pathway in the liver, and the synthesis and functional pathway of neurotransmitters in the brain, thereby improving obesity and cognitive function in rats.

Point 2 Lines 36-40 of the introduction provide brief information without referencing sources.

Response 2 Thank you very much for your comments. I added the corresponding references.

Point 3 Regarding line 90, the notion of administering fat in water seems perplexing as fat is not water-soluble. Additionally, there might be concerns about ensuring each rat receives the appropriate dose, especially when considering the typical method of incorporating fructose and fat into the diet.

Response 3 Thank you very much for your comments. Yes, fat is not soluble in water, I did not express it clearly before. Fat is obtained from diet, only fructose dissolved in water for rat to obtain. In the original, I have added the formulas for each group's diet and the energy supply situation of each diet.

Point 4 Furthermore, it's essential to understand the rationale behind selecting the 20% dose of bean protein.

Response 4 Thank you very much for your comments. I have added the formulas for each group's diet and the energy supply situation of each diet in Table 1 of the document.

Table 1 Composition of Diets

Component

ND

HFFD

HFFD+LP

HFFD+KP

Casein/g

200

200

70

-

L-cystine/g

3

3

3

3

Kidney bean protein/g

-

-

-

200

Corn starch/g

397.5

217.5

347.5

217.5

Maltodextrin 10/g

132

132

132

132

Sucrose/g

100

100

100

100

Cellulose, BW200/g

50

50

50

50

Lard/g

70

250

250

250

Vitamin mix V10037/g

10

10

10

10

Mineral mix S10022G/g

35

35

35

35

Choline bitartrate/g

2.5

2.5

2.5

2.5

Energy supply ratio of each nutrition component

Protein (%)

19

18

6

18

Carbohydrate (%)

65

37

49

37

Fat (%)

15

45

45

45

The abbreviations are as follows: normal diet (ND), high-fat and high-fructose diet (HFFD), high-fat, high-fructose but low protein diet (HFFD+LP) and kidney bean protein diet (HFFD+KP).

Point 5 The duration of treatment for 56 days may raise questions, given the well-established understanding that obesity typically develops over a longer period with an HFFD, typically at least three months.

Response 5 Thank you very much for your comment. The criterion for obesity is that if the weight of the rats in the obese group exceeds that of the rats in the normal group by 10%, we define it as obesity. At the end of the eighth week, the weight of the rats in the obese group had already exceeded that of the rats in the normal group by more than 10%, so we chose to terminate the experiment at the end of the eighth week.

Reviewer 2 Report

Comments and Suggestions for Authors

This study analyzes the effect of Kidney Bean Protein supplementation in dietary induced obese rats. Authors analyze the effects of KP on obesity onset and characteristics as well as on gut microbiota and cognitive status.

In general terms, I believe the work approaches an interesting and relevant topic, the research is very well done, with a suitable introduction, a proper experimental model and data analysis. However, some aspects could be improved.

Comments

1.- The title reflects a causative effect of KP to microbiota and microbiota to cognitive modulation. Results are not sufficient to assure that microbiota changes are the reason for cognitive maintenance, since the content of KP supplementation is not 100% defined. Is it a pure protein supplement or is it a bean extract enriched in protein?. If so, many other secondary metabolites could be involved in the effects, besides the effects on gut microbiota.

To this point, the only proven thing is that KP supplementation suppresses weight gain, improves gut microbiota, and prevents some cognitive deterioration. The interplay between those variables, should be further studied to assess their temporal and causative relationship. Also cited papers about microbiota and cognition , indicate association, not causality. Therefore, lines 59 ,60, 75 and all related statements should be corrected accordingly.

It is tricky to use the term: IMPROVEMENT, since the KP was given during the induction of obesity, not after the onset of obesity or cognitive impairment. So, nothing did improve, but the impairment was prevented. All implications of improvement should be changed to "prevention" or related terms.

2.- In Material and Methods please include how weight and tail length were measured. Please cite any other paper (from other authors) using tail length and weight/tail ratio for nutritional status. The sentence in line 187 is not sufficient to use it.

Also indicate the final KP concentration since the supplement indicates only 85% content, and specify weather some other ingredients might be present.

3.- All figures should be improved, font is too small, specially in Figure 4, light colors are difficult to distinguish and error bars are hard to see.

4.- I highly suggest the performance of a methodology scheme to facilitate understanding and because it is not clear when the water-maze test was performed during the study.

5.- value of n is missing in Figs 3, 4 and 5, was it performed for all 32 rats?

6.- After showing the raw results of gut, brain and liver analysis, a conclusive image should be added to each figure that interprets results between groups. Or, in a 6th figure a complete proposal of relevant metabolites interacting between gut, liver and brain.  

7.- Affirmation of lines 339 to 342, again implicate causality, which has not been proven, just association. 

8.- Discussion about the limitations of the study should be included. For example, that positive results implicate that supplementation should be given during the induction of obesity, and effects of KP to improve all obesity traits, after damage has been established, should be further studied. Also to discuss about the purity of the supplement and the possible effects of other components.

Author Response

Response to Reviewer 2 Comments

Point 1 The title reflects a causative effect of KP to microbiota and microbiota to cognitive modulation. Results are not sufficient to assure that microbiota changes are the reason for cognitive maintenance, since the content of KP supplementation is not 100% defined. Is it a pure protein supplement or is it a bean extract enriched in protein?. If so, many other secondary metabolites could be involved in the effects, besides the effects on gut microbiota.

To this point, the only proven thing is that KP supplementation suppresses weight gain, improves gut microbiota, and prevents some cognitive deterioration. The interplay between those variables, should be further studied to assess their temporal and causative relationship. Also cited papers about microbiota and cognition , indicate association, not causality. Therefore, lines 59 ,60, 75 and all related statements should be corrected accordingly.

It is tricky to use the term: IMPROVEMENT, since the KP was given during the induction of obesity, not after the onset of obesity or cognitive impairment. So, nothing did improve, but the impairment was prevented. All implications of improvement should be changed to "prevention" or related terms.

Response 1 Thank you very much for your comments. I modified the title of the thesis to “Kidney Bean Protein Prevents High-Fat and High-Fructose Diet-Induced Obesity, Cognitive Impairment, and Disruption of Gut Microbiota Composition” and change “improvements” in the paper to prevention.

Point 2 In Material and Methods please include how weight and tail length were measured. Please cite any other paper (from other authors) using tail length and weight/tail ratio for nutritional status. The sentence in line 187 is not sufficient to use it. Also indicate the final KP concentration since the supplement indicates only 85% content, and specify weather some other ingredients might be present.

Response 2 Thank you very much for your comments. In the Animal treatment section, I cited Diana‘s methods for measuring body weight and tail length. I cited the literature on the use of body weight tail length ratio to measure nutritional status in the corresponding places.

Point 3 All figures should be improved, font is too small, specially in Figure 4, light colors are difficult to distinguish and error bars are hard to see.

Response 3 Thank you very much for your comments. I changed the font size of the figure to make it easier to see.

Point 4 I highly suggest the performance of a methodology scheme to facilitate understanding and because it is not clear when the water-maze test was performed during the study.

Response 4 Thank you very much for your comments. I added a methodology scheme to paper.

Point 5 value of n is missing in Figs 3, 4 and 5, was it performed for all 32 rats?

Response 5 Thank you very much for your comments. We collected tissue samples from each group of rats, each group mixed together to complete the index test

Point 6 After showing the raw results of gut, brain and liver analysis, a conclusive image should be added to each figure that interprets results between groups. Or, in a 6th figure a complete proposal of relevant metabolites interacting between gut, liver and brain.

Response 6 Thank you very much for your comments. I have added an analysis of the correlation between gut microbiota, hepatic lipid metabolites, and brain metabolites in the final section.

Point 7 Affirmation of lines 339 to 342, again implicate causality, which has not been proven, just association.

Response 7 Thank you very much for your comments.

Point 8 Discussion about the limitations of the study should be included. For example, that positive results implicate that supplementation should be given during the induction of obesity, and effects of KP to improve all obesity traits, after damage has been established, should be further studied. Also to discuss about the purity of the supplement and the possible effects of other components.

Response 8 We greatly appreciate the reviewers for pointing out the limitations of the conclusions drawn in our paper. We have added a section in the 'Results and Discussion' part to elaborate on these limitations, including: Nonetheless, it is essential to note that in this experimental design, all dietary interventions were conducted concurrently. Therefore, we can only assess the preventive effects of KP on obesity and memory impairments induced by a high-fat and high-lactose diet. In subsequent research, it is necessary to evaluate the treatment effect of KP on established models of obesity and malnutrition; These results suggest that the alterations in gut microbiota induced by KP are likely the primary factor causing changes in serum and brain metabolites, subsequently improving obesity and memory impairments. In further studies, to confirm this conclusion, fecal microbiota transplantation experiments in germ-free mice are imperative. This will also become the focus of our follow-up studies. We hope this can enhance the scientific rigor of our research.

Reviewer 3 Report

Comments and Suggestions for Authors

Dear Authors:

Regarding the manuscript with title “Kidney Bean Protein Suppresses High-Fat and High-Fructose Diet-Induced Obesity and Cognitive Impairment in Rats by Modulation of Gut Microbiota”, I have two major concerns. Also several minor comments were addressed.

Major Comments:

Comment 1:

Manuscripts submitted to Journal Foods must have a chapter of “Discussion”. In this regard, more references must be included.

On Discussion, some examples of themes that authors must discuss are related to the following questions:

- Effectivelly, given the results presented by authors HFFD+KP diet supresses high-fat-high fructose diet-induced obesity and cognitive impairment. However, ND achieved better results than HFFD+KP on several outcomes. This question must be discussed by Authors.

- Tail length gain and its relation with diet.

- No statistically significance on ratio of body weight/tail length; ACE-2 and IGF-1 between HFFD and HFFD+KP

Some information that authors have on the chapter of Results must be transferred to the chapter of Discussion. Some examples:

Lines 182-185: “Shi et al 2020. discovered that kidney bean protein contained functional proteins such as alpha- amylase inhibitors, which had a significant inhibitory effect on the weight gain of obese rats, exhibiting both time and dose-dependency [12, 13].”

Lines 186-188: “Weight-to-length ratio measurements are often used to assess the health status of children, hence, the body weight/tail length ratio of the rats was examined”

Lines 191-194: “This suggested that long-term consumption of  high-fat, high-fructose, and low-protein food can lead not only to obesity but also to malnutrition. Adding kidney bean protein to the diet can significantly mitigate issues of obesity and malnutrition in rats”

Lines 205-206: “in harmony with previous results on tail length and body weight/tail length ratio.”

Comment 2:

I suggest authors to complement Figures with Tables in which authors present: 1) the initial (baseline) and final value (at 8 weeks) of variables; 2) variation between baseline and 8 weeks; 3) p value . It is not clear on Figures the objective values regarding the presented variables.

Minor Comments:

Comment 1:

Line 31-32: I suggest authors to change “among other dietary and lifestyle changes by “among other lifestyle changes”, as lifestyle includes diet.

Comment 2:

I suggest authors to start the chapter of Methods by the subchapter of “Animal treatment”.

Comment 3:

Authors must clarify for the readers who are not familiarized with what is the standard diet AIN-93 M (the percentagem rovision of different nutrients).

Comment 4:

For each diet, authors must present information about the 100% provided by different nutrients. For example on HFFD, authors only presente information regarding 55% of diet (45% from fat and 10% from fructose)

Comment 5:

Lines 94-95: The reference number 10 must be read Burgess et al. (2021) and not Brown et al. (2021), as Burgess was the first author. Why authors based the experimental design on the study of Burgess et al. (2021)?

Comment 6:

Line 179: “Rats were fed with four different diets for 8 weeks”. This information must be transferred for the subchapter “Animal treatment”

Comment 7:

Lines 181-182: “This revealed that the inclusion of kidney bean protein in the rat diet suppressed weight gain induced by a high-fat high-fructose diet.”. Authors must state if the results were statistically significant or not. From what is stated on Figure 1, results were statisticaaly significant between HFFD and the other 3 diets. Thus, authors must not individualize only for HFFD+KP diet. Besides, when compared to HFFD diet, the ND was the one that achieved a bigger weight reduction

Comment 8:

Lines 185-186: “The tail lengths were reduced respectively by 2.8%, 14% and 3.7% (Figure 1C).” On figure 1c, authors presented results of 4 diets. The previous percentages referred to which diets. I supposed that authors want to follow the same mechanism they applied on line 180, but they have to refer that compared to HFFD, rats fed with ND, HFFD+LP and HFFD+KP diets reduced respectivelly 2.8%, 14% and 3.7%..

Another 2 important considerations:

1. The tail lengths were not reduced as stated on line 185.. What readers can see on figure 1c is that the tail lengths increases.

2. On Figure 1c, authors must change the word “lenth” by “length”

Comments on the Quality of English Language

Some minor changes were suggested on Comments.

Author Response

Point 1 Manuscripts submitted to Journal Foods must have a chapter of “Discussion”. In this regard, more references must be included. On Discussion, some examples of themes that authors must discuss are related to the following questions:

- Effectivelly, given the results presented by authors HFFD+KP diet supresses high-fat-high fructose diet-induced obesity and cognitive impairment. However, ND achieved better results than HFFD+KP on several outcomes. This question must be discussed by Authors.

- Tail length gain and its relation with diet.

- No statistically significance on ratio of body weight/tail length; ACE-2 and IGF-1 between HFFD and HFFD+KP

Some information that authors have on the chapter of Results must be transferred to the chapter of Discussion. Some examples:

Lines 182-185: “Shi et al 2020. discovered that kidney bean protein contained functional proteins such as alpha- amylase inhibitors, which had a significant inhibitory effect on the weight gain of obese rats, exhibiting both time and dose-dependency [12, 13].”

Lines 186-188: “Weight-to-length ratio measurements are often used to assess the health status of children, hence, the body weight/tail length ratio of the rats was examined”

Lines 191-194: “This suggested that long-term consumption of high-fat, high-fructose, and low-protein food can lead not only to obesity but also to malnutrition. Adding kidney bean protein to the diet can significantly mitigate issues of obesity and malnutrition in rats”

Lines 205-206: “in harmony with previous results on tail length and body weight/tail length ratio.”

Response 1 Thank you very much for your comments. I have added a discussion section to the results section, where I discuss the content of each section separately.

Point 2 I suggest authors to complement Figures with Tables in which authors present: 1) the initial (baseline) and final value (at 8 weeks) of variables; 2) variation between baseline and 8 weeks; 3) p value . It is not clear on Figures the objective values regarding the presented variables.

Response 2 Thank you very much for your comments.I turned Figure 1 in the paper into a table and added a significance analysis

Minor Comments:

Point 1 Line 31-32: I suggest authors to change “among other dietary and lifestyle changes by “among other lifestyle changes”, as lifestyle includes diet.

Response 1 Thank you very much for your comments. I changed “among other dietary and lifestyle changes” to “among other lifestyle changes”.                           

Point 2: I suggest authors to start the chapter of Methods by the subchapter of “Animal treatment”.

Response 2 Thank you very much for your comments. In this article I set Animal treatment as the first part of chapter of Methods.

Point 3 Authors must clarify for the readers who are not familiarized with what is the standard diet AIN-93 M (the percentagem rovision of different nutrients).

Response 3 Thank you very much for your comments. I have added the formulas for each group's diet and the energy supply situation of each diet in Table 1 of the document.

Table 1 Composition of Diets

Component

ND

HFFD

HFFD+LP

HFFD+KP

Casein/g

200

200

70

-

L-cystine/g

3

3

3

3

Kidney bean protein/g

-

-

-

200

Corn starch/g

397.5

217.5

347.5

217.5

Maltodextrin 10/g

132

132

132

132

Sucrose/g

100

100

100

100

Cellulose, BW200/g

50

50

50

50

Lard/g

70

250

250

250

Vitamin mix V10037/g

10

10

10

10

Mineral mix S10022G/g

35

35

35

35

Choline bitartrate/g

2.5

2.5

2.5

2.5

Energy supply ratio of each nutrition component

Protein (%)

19

18

6

18

Carbohydrate (%)

65

37

49

37

Fat (%)

15

45

45

45

The abbreviations are as follows: normal diet (ND), high-fat and high-fructose diet (HFFD), high-fat, high-fructose but low protein diet (HFFD+LP) and kidney bean protein diet (HFFD+KP).

Point 4 For each diet, authors must present information about the 100% provided by different nutrients. For example on HFFD, authors only presente information regarding 55% of diet (45% from fat and 10% from fructose)

Response 4 Thank you very much for your comments. Thank you very much for your comments. I have added the formulas for each group's diet and the energy supply situation of each diet in Table 1 of the document.

Point 5 Lines 94-95: The reference number 10 must be read Burgess et al. (2021) and not Brown et al. (2021), as Burgess was the first author. Why authors based the experimental design on the study of Burgess et al. (2021)?

Response 5 Thank you very much for your comments. I revised the literature cited.

Point 6 Line 179: “Rats were fed with four different diets for 8 weeks”. This information must be transferred for the subchapter “Animal treatment”

Response 6 Thank you very much for your comments. I added the corresponding content in the methods section, as follows. All rats were housed under controlled conditions at 24 °C, with a 12-hour light/dark cycle. Body weight and tail length were measured every two weeks for a total feeding period of eight weeks

Point 7 Lines 181-182: “This revealed that the inclusion of kidney bean protein in the rat diet suppressed weight gain induced by a high-fat high-fructose diet.”. Authors must state if the results were statistically significant or not. From what is stated on Figure 1, results were statisticaaly significant between HFFD and the other 3 diets. Thus, authors must not individualize only for HFFD+KP diet. Besides, when compared to HFFD diet, the ND was the one that achieved a bigger weight reduction

Response 7 Thank you very much for your comments. In the chapter of Effect of supplementary kidney bean protein on the growth profile of obese rats, I discussed the differences in body weight between ND and HFFD + LP, HFFD + KP. The details are as follows. It is noteworthy that at the eighth week, compared to the ND group, there was no significant difference in the body weight of rats in the HFFD+LP and HFFD+KP groups. Although the HFFD+LP diet contained more fat and sugar, the content of protein in the diet was reduced. The reduction in protein intake led to malnutrition, limiting the growth of rats. The reason why the body weight of the rats in the HFFD+KP group showed no significant difference with the ND group could be that the kidney bean protein contains functional proteins such as alpha-amylase inhibitors, which had a significant inhibitory effect on the weight gain of obese rats

Point 8 Lines 185-186: “The tail lengths were reduced respectively by 2.8%, 14% and 3.7% (Figure 1C).” On figure 1c, authors presented results of 4 diets. The previous percentages referred to which diets. I supposed that authors want to follow the same mechanism they applied on line 180, but they have to refer that compared to HFFD, rats fed with ND, HFFD+LP and HFFD+KP diets reduced respectivelly 2.8%, 14% and 3.7%.

Response 8 Thank you very much for your comments. I will modify this sentence as at 8-week, compared to the HFFD+LP group, rats fed with ND, HFFD and HFFD+KP diets showed tail length increased by 11.5%, 14.0% and 10.6% respectively (p<0.05)

Another 2 important considerations:

Point 1 The tail lengths were not reduced as stated on line 185.. What readers can see on figure 1c is that the tail lengths increases.

Point 1 Thank you very much for your comments. I made a mistake in my previous paper.

Point 2 On Figure 1c, authors must change the word “lenth” by “length”

Response 2 Thank you very much for your comments. I turned the graph in the paper into a table.

Round 2

Reviewer 1 Report

Comments and Suggestions for Authors

Dear Editor's

Thank you for the opportunity to revise our manuscript. We have carefully addressed the reviewers' comments and implemented the requested changes. In response to your request, we have made additional revisions to further enhance the clarity and comprehensiveness of the manuscript.

Specifically, for Tables 1, 2, and 3, we have included the statistical significance of variables "a" and "b" as per your suggestion. Additionally, we have formatted these statistical significances in superscript format for better readability and presentation.

Furthermore, for Figures 3A and 3B, we have added information regarding the significance of variation between groups, as requested.

Thank you 

Author Response

Point 1 Thank you for the opportunity to revise our manuscript. We have carefully addressed the reviewers' comments and implemented the requested changes. In response to your request, we have made additional revisions to further enhance the clarity and comprehensiveness of the manuscript.

Specifically, for Tables 1, 2, and 3, we have included the statistical significance of variables "a" and "b" as per your suggestion. Additionally, we have formatted these statistical significances in superscript format for better readability and presentation.

Furthermore, for Figures 3A and 3B, we have added information regarding the significance of variation between groups, as reques

Response 1 Thank you very much for your comments. I add significance of variation in Table 2 and Table 3. In Fig.3A, there was no significant difference in the four groups of ND, HFFD, HFFD + LP and HFFD + KP. In Figure 3B, I added significance of variation.

Reviewer 3 Report

Comments and Suggestions for Authors

Dear Authors.

The manuscript has improved its quality after the changes.

I still have two comments that I kindly ask to be clarified. 

Comment 1:

On Table 2, what means letters a, b and c? This must be clarified in the legend of the Figure. Also, what means n=8? 

Comment 2:

On the first version of the manuscript, authors refer that the experimental design was based on that of Brown et al. (2021)

Burgess NM.; Kelso W.; Malpas CB.; Winton-Brown T.; Fazio T.; Panetta J.; De Jong G.; Neath J.; Atherton S.; Velakoulis D.; et 377 al. The Effect of Improved Dietary Control on Cognitive and Psychiatric Functioning in Adults with Phenylketonuria: the 378 ReDAPT Study. Orphanet Journal of Rare Diseases. 2021; 16 (1)

On the revised version, authors refer that the experimental design was based on that of Brown et al. (2021). The reference is different from that stated on the first version and the year of publication is 2015 and not 2021.

Brown EM.; Wlodarska M.; Willing BP.; Vonaesch P.; Han J.; Reynolds LA.; Arrieta MC.; Uhrig M.; Scholz R.; Partida O.; et al. 429 Diet and specific microbial exposure trigger features of environmental enteropathy in a novel murine model. Nat Commun. 2015; 430 6: 7806

I kindly ask authors to

- refer clearly which reference is the basis of the experimental design and

- why authors based the experimental design in that reference?

Author Response

Point 1 On Table 2, what means letters a, b and c? This must be clarified in the legend of the Figure. Also, what means n=8?

Response 1 Thank you very much for your comments. I explain the meaning of a, b and c in the comments of Table 2 and Table 3. N = 8 represents the index information of eight rats was collected.

Table 2 Rat body weight, tail length, body weight tail length ratio within eight weeks

Group

0 week

2 week

4 week

6 week

8 week

Body
weight(g)

ND

77.84±3.47b

172.65±8.57a

236.93±5.65b

317.04±8.10c

391.03±7.95b

HFFD

85.11±4.60a

180.76±6.87a

250.64±10.22a

344.14±8.8a

430.28±14.92a

HFFD+LP

84.46±6.58ab

176.75±10.74a

244.61±9.90ab

337.14±12.45ab

403.09±11.69b

HFFD+KP

85.55±5.60a

177.49±7.27a

243.94±7.61ab

328.60±8.53bc

397.18±8.11b

Tail length(cm)

ND

9.71±0.57a

13.18±0.84a

14.80±0.75b

16.31±0.69ab

18.23±0.80a

HFFD

10.19±0.57a

13.49±1.20a

15.33±1.15a

17.03±1.02a

18.75±1.08a

HFFD+LP

10.39±0.20a

13.01±0.77a

14.08±0.87ab

15.40±0.73b

16.13±0.73b

HFFD+KP

11.45±3.39a

13.34±0.72a

14.95±0.67ab

16.48±0.59a

18.05±0.84a

Body
weight/Tail length

ND

8.03±0.40a

13.12±0.44a

16.04±0.67b

19.46±0.65b

21.48±0.73c

HFFD

8.36±0.26a

13.47±0.81a

16.41±0.83b

20.27±0.98b

23.00±1.05a

HFFD+LP

8.13±0.55a

13.60±0.69a

17.42±0.79a

21.93±1.07a

25.03±0.95b

HFFD+KP

7.89±1.51a

13.30±0.44a

16.23±0.56b

19.86±0.50b

21.96±0.98bc

Abbreviation: ND, normal diet; (2) HFFD, high-fat and high-fructose diet; (3) HFFD+LP, high-fat, high-fructose but low protein diet; (4) HFFD+KP, high-fat, high-fructose but kidney bean protein diet. The results were expressed as mean±SD (n=8) and different letters in the same column indicate significant differences (p<0.05).

Table 3 The escape latency of rats in four days

Group

1

2

3

4

ND

41.00±1.41d

30.50±1.50c

26.88±1.27b

24.50±0.87c

HFFD

46.13±1.83b

33.13±1.45b

28.00±1.12b

28.50±1.66b

HFFD+LP

59.00±0.71a

52.00±2.18a

41.25±2.11a

41.00±1.41a

HFFD+KP

43.5±1.87c

31.13±2.03bc

27.38±1.87b

26.00±1.22c

The results were expressed as mean ± SD (n=8) and different letters in the same column indicate significant differences (p<0.05).

Point 2 On the first version of the manuscript, authors refer that the experimental design was based on that of Brown et al. (2021)

Burgess NM.; Kelso W.; Malpas CB.; Winton-Brown T.; Fazio T.; Panetta J.; De Jong G.; Neath J.; Atherton S.; Velakoulis D.; et 377 al. The Effect of Improved Dietary Control on Cognitive and Psychiatric Functioning in Adults with Phenylketonuria: the 378 ReDAPT Study. Orphanet Journal of Rare Diseases. 2021; 16 (1)

On the revised version, authors refer that the experimental design was based on that of Brown et al. (2021). The reference is different from that stated on the first version and the year of publication is 2015 and not 2021.

Brown EM.; Wlodarska M.; Willing BP.; Vonaesch P.; Han J.; Reynolds LA.; Arrieta MC.; Uhrig M.; Scholz R.; Partida O.; et al. 429 Diet and specific microbial exposure trigger features of environmental enteropathy in a novel murine model. Nat Commun. 2015; 430 6: 7806

I kindly ask authors to

- refer clearly which reference is the basis of the experimental design and

- why authors based the experimental design in that reference?

Response 2 Thank you very much for your comments. I refer to Brown published in Nature Communications in 2015. The title of this paper is Diet and specific microbial exposure trigger features of environmental enteropathy in a novel murine model. After reading this article, I think the experimental method of this article will help inspire me to design the experiment. I changed Brown et al. 2021 to Brown et al. 2015 in line 88.
